# Fracture Strength and Failure Modes of Endodontically Treated Premolars Restored with Compact and Hollow Composite Posts Subjected to Cyclic Fatigue

**DOI:** 10.3390/ma15031141

**Published:** 2022-02-01

**Authors:** Vito Gallicchio, Vincenzo Lodato, Roberto De Santis, Sandro Rengo

**Affiliations:** 1Department of Neurosciences, Reproductive and Odontostomatological Sciences, University “Federico II” of Naples, Via S. Pansini 5, 80131 Naples, Italy; vito.gallicchio@unina.it (V.G.); vincenzo.lodato@unina.it (V.L.); sanrengo@unina.it (S.R.); 2Institute of Polymers, Composites and Biomaterials—National Research Council of Italy, V.le J.F. Kennedy 54—Mostra d’Oltremare Pad. 20, 80125 Naples, Italy

**Keywords:** fiber reinforced composite, particulate composite, endodontic post, fatigue, bending, fracture strength

## Abstract

Physical and mechanical properties of continuous carbon or glass fiber reinforced endodontic posts are relevant to increase the retention and resistance of the tooth-restoration system. Hollow posts have been recently designed for delivering the luting cement through the post hole, thus enhancing the post-dentin interface by reducing the risk of air bubbles formation. Methods: Three type of endodontic posts, a carbon fiber hollow post, a glass fiber hollow post and a compact glass fiber post were investigated. Mechanical properties of these posts were assessed through bending tests. Teeth were subjected to fatigue cycling and the strength of restored teeth was detected through static tests. Failure modes were investigated through optical and scanning electron microscopy. Results show that composite posts increase the mechanical stability by more than 100% compared to premolars restored with particulate composite. Carbon fiber posts retain the highest strength (1467 N ± 304 N) among the investigated post and core restoration, but an unfavorable type of fracture has been observed, preventing the tooth re-treatment. Instead, more compliant posts (i.e., glass fiber reinforced composite, providing a strength of 1336 N ± 221 N), show a favorable mode of fracture that allows the re-treatment of teeth in the case that failure occurs. Glass fiber hollow posts show a good trade-off between strength and a favorable type of fracture.

## 1. Introduction

The main goal of conservative dentistry is to preserve teeth and to reduce extractions as much as possible. Endodontic treatment is the gold standard to prevent decayed teeth from extraction, although endodontically treated teeth are prone to fracture [1,2]. The fracture can be caused by the pulpar roof loss produced by endodontic access, and even carious processes can remove the remaining tooth structure, thereby weaking it. First maxillary premolars have a high incidence of fractures due to the small root diameter. Premolars are subjected to higher lateral forces during mastication compared to the anterior teeth. Such elements often require cusp coverage and indirect reconstruction techniques (overlays, crowns), while direct restoration is intended for cases in which vestibular and palatal cusps are well represented and at least one marginal ridge is present [3].

Endodontically treated premolars show a risk of mechanical failure higher than teeth with pulp vitality, and for this reason a variety of polymer-based composite materials including particles and fiber reinforced composites have been investigated for increasing the biomechanical stability of endodontic treated premolars [4]. Physical and mechanical properties of post-endodontic restorations are relevant to increase the retention and resistance of the tooth restoration system. Several studies have demonstrated that the use of continuous fiber reinforced endodontic posts increases the material build up core retention, depending on the rigidity of the material of post core systems [4,5,6]. A property of great importance for composite materials is strength, which provides information on the amount of force the material can withstand and also the ability to react to external loads. Generally, when two or more materials with different rigidities are coupled, the stress is not uniformly distributed, and the major stress is being transferred from the rigid material to the compliant one [7]. Strength and rigidity are the key to balance the stress distribution between materials, and in the last years research has focused on the development of fiber posts with a trade-off between rigidity and compliance [8]. In particular, the goal of fiber reinforced posts is to achieve an elastic modulus similar to that of dentin, allowing for a mechanically homogeneous system [9]. From the late 1990s carbon fiber and glass fiber posts have been available highly rigid metal posts. These posts present an elastic modulus similar to that of dentin, and the stress transferred to the nearest tissues is distributed uniformly, thus preventing root fractures frequently observed in metal post restored teeth [10].

Pre-fabricated fiber reinforced posts need to be cemented into the root canal in situ, therefore post space preparation into the dental root through drills and reamers and the use of a resin luting agent are implemented for anchoring the endodontic post [5]. The bond strength between the resin luting agent and the root canal surface is influenced by the polymerization degree of resin cement in the post cavity. Specifically, a lower polymerization degree can be found at the bottom of the cavity due to the decrease of light cure intensity [11]. Lately, self-adhesive resin cement characterized by a dual-cure mechanism and no dentin pre-treatment requirement has been introduced on the dental market [12]. The presence of two residual marginal ridges significantly improves long term success of post-endodontic restorations [13]. A prospective in vivo study showed similar three-year survival rates between endodontically treated premolars restored with fiber posts and direct composite resin and complete restorations with metal ceramic crowns [14].

A new type of endodontic composite post has been recently developed and it is available on the dental market, the hollow fiber post [15]. Fiber reinforced hollow posts represent an advancement over compact fiber posts. In particular, the main advantage of hollow posts is the reverse extrusion of the luting cement involving an application of the cement from the apex to the crown of the tooth to be restored. Thanks to this technique, the cementation process is optimized by avoiding the entrapment of air bubbles and the stability of the hollow post is improved by a homogeneous distribution of the cement in the root cavity hosting the post itself [15]. However, little is known about the mechanical performance of hollow fiber reinforced endodontic posts and their capability to restore teeth.

Currently, there is insufficient evidence of fracture resistance and fracture patterns in endodontically treated teeth to define the proper use of fiber posts, especially when teeth are subjected to thermomechanical aging and simulation of masticatory forces as in the oral environment [16]. It has been shown that in vitro the reduced stiffness of certain fiber-reinforced posts may be beneficial in preventing catastrophic root fractures [17]. However, it is not clear whether fiber-reinforced posts can actually provide adequate support for a core. The deformation of a fiber-reinforced post may result in greater stress on the composite core, causing premature failure of the core restoration [18]. This problem is of particular clinical interest in the cases where little or no coronal tooth structure remains.

Endodontic composite posts mainly consist of straight fibers (i.e., glass or carbon fibers) embedded into an epoxy matrix. The stability of the fiber-matrix interface in a wet environment largely depends on the quality of this interface, and it is reported that, in chloride solution, the interfacial damage between fibres and matrix is delayed as a smooth-surfaced composite is used [19]. On the other hand, debonding of the fiber/resin interface in rods made of carbon/glass fiber reinforced epoxy has been observed, and it has been ascribed to water diffusion that hydrolyses and plasticizes the resin matrix, weakening the bond between fiber and resin [20]. Therefore, it is very important to replicate the complex oral environment for assessing in vitro the biomechanical behaviour of applied composite endodontic posts. Little is known on the in vitro behavior of premolars restored with endodontic posts stressed through cyclic loading in a wet environment at 37 °C.

The aim of the current study was to evaluate the fracture resistance and fracture patterns of endodontically treated maxillary premolars subjected to mesial-occlusal-distal (MOD) cavities restored using a resin composite core with different types of fiber posts under cyclic loading in a wet environment at 37 °C.

## 2. Materials and Methods

### 2.1. Endodontic Posts and Cement Selection

Three endodontic posts, a hollow carbon fiber post (HCP), a hollow glass fiber post (HGP), a compact glass fiber post (GP), and the self-etching and self-adhesive dual cement Maxcem Elite Chroma (MEC) were used. Details of the selected posts and cements are reported in Table 1. Figure 1a shows the three types of endodontic posts used in our investigation.

### 2.2. Three-Point Bending Tests

Three-point bending tests were performed on each sample of endodontic posts using the Instron 5566 dynamometer (Instron Ltd., High Wycombe, UK) equipped with a load cell of 100 N. The span (L) was set at 13 mm and bending tests were carried out at a speed of 1 mm/min. Each sample consisted of 10 specimens.

The Maxcem Elite Chroma cement was tested in the same fashion of endodontic posts. Dental cement specimen bars were photo-cured in Teflon moulds as previously described [11]. Briefly, the cement was injected into a prismatic cavity mould having a cross section of 1 × 1 mm^2^, and a Mylar strip was used to cover the mould and to avoid the composite oxidation during the curing process. A Swiss Master Light (EMS) curing unit at an intensity of 1000 mW/cm^2^ was employed. The curing unit and the mould containing the cement were fixed onto a modified 3D CAD/CAM system [21] in order to provide a continuous curing process along the whole length of the specimen bar (15 mm), and the process was performed within 20 s of light exposure.

A sample consisting of ten specimens of hollow carbon fiber post filled with Maxcem Elite Chroma cement (HCP + MEC) was also considered. The cement was injected into the cavity of the hollow carbon fiber post. A curing process for 20 s was considered, and specimens were stored in a dry and dark environment for 72 h in order to allow the polymerization of the dual cement. A sample consisting of ten specimens of hollow glass fiber post filled with Maxcem Elite Chroma cement (HGP + MEC) was also prepared in a similar fashion.

The bending behavior of HCP, HCP + MEC, GP, HGP, HGP + MEC and MEC was described through load-displacements curves. The steepness of the linear portion of each curve was computed through the best linear curve fit using the Kaleidagraph software (Synergy Software, Reading, PA, USA).

The Young’s modulus (E) of each material was computed using the following equation:(1)$$\frac{\mathrm{dP}}{\mathrm{dy}}=\frac{48\;E\;I}{L^{3}}$$
where P is the applied load, y the displacement of the middle-span cross-section, *L* is the span and I is the second moment of area. The product EI is known as bending stiffness.

The second moment of area of compact posts (i.e., GP), hollow posts (i.e., HCP and HGP) and cement (MEC) were computed according to the following equations:(2)$$I=\frac{\pi }{4}\;R^{4};\;I=\frac{\pi }{4}\;(R^{4}-r^{4});\;I=\frac{b^{4}}{12}$$
where *R*, *r* and *b* were the external post’s radius, the internal post’s radius and the thickness of the cement bar, respectively.

For hollow posts filled with cement (i.e., HCP + MEC and HGP + MEC), the following equation was used to verify the experimental linear behavior with the theoretical response once the Young’s moduli of the single components were experimentally computed:(3)$$\frac{\mathrm{dP}}{\mathrm{dy}}=\frac{48\;}{L^{3}}\left(E_{e\;}I_{e\;}+E_{i\;}I_{i\;}\right)$$
where *E_e_* and *I_e_* are the Young’s modulus and the second moment of area of the external composite shell, respectively. While *E_i_* and *I_i_* represent the Young’s modulus and the second moment of internal cement core, respectively.

On the other hand, the axial stiffness of each sample is obtained by the product of the area of the specimen cross-section and the Young’s modulus. For HCP + MEC and HGP + MEC the axial stiffness is given by Ee·Ae + Ei·Ai, where Ae and Ai are the cross-section of the external shell and the internal core, respectively.

### 2.3. Maxillary First Premolar Selection

Fifty maxillary first premolars were selected. Teeth were extracted for orthodontic reasons. The study was approved by the Ethics Committee of the University of Naples Federico II, with protocol number 137 2017. Inclusion criteria were no carious tissue, similar crowns and roots sizes, two root canals, no abfractions, no cracks, no erosions. Teeth were placed in 5% NaOCl solution for 5 min and stored in physiological solution at room temperature to prevent dehydration.

Three endodontic posts, a hollow carbon fiber post, a hollow glass fiber post and a compact glass fiber post, were used. Teeth restored with posts were compared with control teeth restored using a resin-based composite and with healthy, sound teeth. Teeth were subjected to fatigue loading and, finally, compression tests to evaluate the fracture resistance and fracture patterns.

### 2.4. Root Canal Treatment and Obturation

Specimens were subjected to endodontic treatment except for the control group (healthy teeth). Cavity access was prepared with a diamond spheric bur mounted on the turbine Fona8080 (Fonadental, Assago, Italy). Canals were scouted at work length using k-file 10 (Kerr Corporation, Orange, CA, USA). The glide path was performed using a pro-glider (Maillefer, Ballaigues, Switzerland). Canals were instrumented by using the crown down technique with rotating files ProTaper next x1-x2 taper 0.04 (Maillefer, Ballaigues, Switzerland). After each file change, canals were irrigated with 2.5% NaOCl solution. Root canals were dried with size 25 paper points and obturated with gutta-percha cones using a single-cone technique. The samples were stored at 37 °C and 100% humidity. All teeth were manually prepared by an experienced operator. Samples were randomly divided into five groups (n = 10) as shown below, and different fiber posts were employed for the restoration.

Group 1

Teeth in this group served as the control group (n = 10) and they were not subjected to any procedure.

Group 2

Class 2 mesio-occlusal-distal cavities were prepared with gingival margin at the cement-enamel junction level. The thickness of the bucco-lingual cavity was 3 mm measured with a digital caliper (Mitutoyo, Takatsuku, Kawasaki, Japan). Teeth were then subjected to an adhesive procedure by acid etching of enamel and dentin for 30 s and 15 s, respectively, using 37% phosphoric acid (Gerhò, Bolzano, Italy) and subsequentially 5 s of rinsing and drying; finally, the application of the Optibond SE (Kerr Corporation) adhesive system and photo-curing with the Swiss Master Light (EMS) curing unit at an intensity of 1000 mW/cm^2^ [22] SonicFill was directly dispensed into dental cavities using the Sonicfill (Kerr Corporation) handpiece.

Group 3

Palatal Root canals were subjected to post space preparation. A Tech21 compact glass fiber post (Isasaan, Como, Italy) was inserted in post-endodontic restoration. Post housing was prepared in the palatal canal using the Gates Glidden bur n. 4, and then fiber posts were inserted. The self-etching and self-adhesive dual cement Maxcem Elite Chroma (Kerr corporation) were used for luting fiber posts, according to manufacturer’s instructions. Tech21 posts were immersed in the cement and inserted into the canal. Maxcem was placed into the post spaces using a Lentulo spiral (Dentsply Sirona, Bensheim, Germany). Light curing was performed using the Swiss Master Light (EMS, Domat, Switzerland) curing unit at 800 mW/cm^2^ for 40 s. After post cementation, teeth were restored with direct restoration using a composite. Restorations were done as in Group 2.

Group 4

Post space and restorations were done as in Group 3, except for the cementation and type of post. A Techole (Isasaan, Como, Italy) hollow glass fiber post was inserted in the post-space restoration. While Tech21 posts were immersed in the cement and inserted into the canal, Techole posts were inserted into the canal and cement was then applied through the central hole, thus flowing through the entire Techole post up to its extremity. Therefore, by using these posts it is no longer necessary to first fill the canal with the cement and then insert the post. Instead, everything is accomplished in one single step, during which the post is simultaneously an instrument which is able to guide the cement into the canal. After post cementation, teeth were restored with direct restoration using a composite as was done in previous groups.

Group 5

With regard to this group, the same procedures as in Group 4 were applied. The only difference consisted in the use of the Techole hollow carbon fiber post.

### 2.5. Fatigue Loading and Fracture Test

Premolars were cemented in aluminum cylinders (D = 16 mm) using a low temperature self-curing acrylic resin 2 mm below their cementum-enamel junction (CEJ) to simulate crestal bone.

Next, specimens were subjected to fatigue stress cyclic loading, in a water environment at a constant temperature of 37 °C, for one million cycles with a sinusoidal variable loading in the range 10–100 N at a frequency of 2 Hz. Figure 1b shows the cyclic loading condition; the load is applied through a horizontal stainless cylinder (with a diameter of 6 mm) placed between the two cusps of each premolar. The dynamically applied vertical load can be decomposed in a vertical compressive load and a horizontal bending load. Therefore, dental specimens undergo a complex fatigue stress cyclic loading consisting of the combination of compressive and bending loading.

After fatigue cycles, teeth were subjected to static mechanical tests using the Instron 5566 testing machine at a speed of 1 mm/min. The dynamometer compliance was experimentally measured and the compression stiffness was evaluated through the steepness of the stress–strain curve in the elastic region.

Finally, data were statistically analyzed using two-way ANOVA followed by Tukey’s test at a critical value of 0.05. The mean and standard deviation (SD) of fracture resistance in each group was calculated.

### 2.6. Optical and Scanning Electron Microscopy

Optical and scanning electron microscopy (SEM) were used to study the fracture behavior of teeth that underwent cyclic fatigue and static tests to failure. The optical microscope Motic AE21 (Motic Ltd., Kowloon, Hong Kong) equipped with a Nikon D3200 camera was implemented to investigate the type of fracture pattern at a macroscale level. Fracture propagating from the crown to the level below the CEJ was considered unfavorable, as they cannot be restored later. Fractures which edge above the CEJ with a minimum distance of 1 mm are classified as positive and favorable as they can be easily restored later [10,23].

SEM was performed using the Inspect S-50 (Thermo Fisher Scientific, Hillsboro, OR, USA) and fracture details were investigated at a microscale level. In order to accommodate specimens on the microscope stub, tooth crowns were sectioned using the microtome IsoMet (Buehler Ltd., Lake Bluff, IL, USA) at a speed of 60 rpm. Finally, specimens were metallized by applying an ultra-thin coating of electrically conducting metal (Gold) to improve the imaging of samples.

Adhesive fractures occurring at the adhesive interfaces (i.e., composite-dentin and post-composite interfaces) and cohesive fractures (i.e., dentin and composite fractures) were distinguished.

## 3. Results

Figure 2 reports the mechanical behavior in bending of posts and cements. An initial linear portion detecting the elastic behavior of each composite post and cement can be distinguished. The steepness of each curve in the linear region (dP/dy) was computed through the linear best fit, and results are reported in Table 2. The steepness of MEC is significantly lower (*p* < 0.01) than those of all the other samples, while the steepness of HCP and HCP + MEC is significantly higher (*p* < 0.01) than those detected for all the other samples. However, no significant difference was observed between the steepness of HCP + MEC and HCP, suggesting that the contribution of the composite cement (i.e., MEC) filling the cavity of the hollow carbon fiber post can be neglected. Similarly, no significant difference was found between the steepness of HGP and HGP + MEC, suggesting that the contribution of the composite cement (i.e., MEC) filling the cavity of the hollow glass fiber post can be neglected. Moreover, this contribution does not allow us to reach steepness values typical of compact glass posts (GP), as the steepness of GP is significantly higher (*p* < 0.01) than that of HGP + MEC.

While composite hollow posts filled with cement (i.e., HCP + MEC and HGP + MEC), compact glass posts (GC) and the particulate composite cement showed a well-defined break point characterized by a crash fracture, hollow posts (i.e., HCP and HGP) did not show a well-defined break point, as these specimens underwent an instability behavior of the mid-span cross section. This instability is shown by the complex load-displacement behavior following the maximum point of HCP and HGP.

The maximum load measured for HCP + MEC (50.34 ± 1.78 N) was significantly higher than that of HCP (31.57 ± 1.52 N); the composite cement filling the cavity of the hollow carbon fiber post increased the bending strength of about 59%. A similar result concerning the reinforcing effect of MEC was observed for the hollow glass fiber post, and an increase of the bending strength of about 51% was recorded.

Table 2 reports the Young’s modulus of HCP, HGP, GP and MEC computed according to Equation (1). The rigidity of HCP is significantly higher (*p* < 0.01) than the other endodontic posts. A significantly lower rigidity (*p* < 0.01) was observed for the composite cement.

For HCP, HGP, GP and MEC, the axial stiffness is given by the product between the Young’s modulus and the first moment of area, and these values are reported in Table 2. Similarly, the bending stiffness is given by the product between the Young’s modulus and the second moment of area, and these values are also reported in Table 2.

The axial stiffness and the bending stiffness of hollow fiber reinforced posts filled with cement are computed by adding the contribution to the stiffness of the external shell and internal core (Table 2).

For HGP + MEC, a theoretical steepness of 84.33 N/mm was calculated through Equation (3) by knowing the Young’s moduli of the single components (i.e., the external fiber reinforced shell and the composite core); this value is consistent with the experimental value reported in Table 2 for HGP + MEC. Similarly, a theoretical steepness of 128.68 N/mm was computed for HCP + MEC by using Equation (3). This value is consistent with the experimental value reported in Table 2 for HCP + MEC.

Figure 3 reports the mechanical behavior after fatigue cycling for maxillary premolars (Group 1), premolars having an MOD restored with composite (Group 2), and premolars having an MOD cavity restored with the investigated endodontic posts (Groups 3 to 5).

Table 3 reports the values of mechanical strength for premolars and restored MOD cavities of premolars recorded after fatigue.

The strength of sound teeth (Group 1) was significantly higher (*p* < 0.01) than all the other groups. Premolars having an MOD restored with composite (Group 2) showed a mechanical strength significantly lower (*p* < 0.01) than all the other groups. No significant difference was observed in the strength of maxillary premolars restored with the different endodontic posts (Groups 3 to 5).

The failure mode of sound premolars and premolars having an MOD cavity restored with composite or with the different endodontic posts is shown in Figure 4.

Fracture patterns of the specimens were evaluated using a digital microscope. Based on the failure mode, fracture types were classified into favorable and unfavorable depending on the position between the CEJ and the fracture surface lower edge (Figure 5). Fracture edges above the CEJ and with a minimum distance of 1 mm are defined as positive and favorable and can be easily restored later. Furthermore, fractures below the CEJ that exceed 1 mm distance are defined as negative and unfavorable as they cannot be restored later. MOD cavities restored through composite (Group 2) or through the more compliant glass fiber post (Group 4) showed a favorable type of fracture. Instead, MOD cavities restored through the more rigid carbon fiber post showed an unfavorable type of fracture.

High resolution images of the fractured specimens obtained through SEM imaging are reported in Figure 6.

The control group (Group 1) showed a cohesive fracture (Figure 6a) occurring through the dentine. Premolars restored with the composite material (Group 2) mainly displayed an adhesive type of fracture (Figure 6b) occurring at the composite-dentine interface propagating through the dentine above the CEJ. Teeth restored through fiber posts (Groups 3, 4 and 5) showed a mixed type of fracture (Figure 6c–e) involving the adhesive interface between the fiber post and the cement, and a cohesive interface occurring in both the luting cement and dentine.

## 4. Discussion

In a dental restoration involving composites, materials with different properties and different elastic modulus meet at the adhesive interface layer that is the weaker area of the restoration, as debonding is recognised as the main cause of the restorative failure [24]. Mechanical properties of fiber posts are also relevant for the success of endodontic restorations, since highly rigid endodontic posts transfer the chewing stress apically and stress concentration may lead to restoration failure and root fracture [6]. Continuous fiber reinforced polymers offer the possibility to tailor the stiffness through a material design, and functionally graded composites represent an elegant strategy for designing advanced endodontic posts [25]. Inspired by natural design, the possibility to locate the fiber reinforcement far from the neutral axis represents another strategy for tailoring mechanical properties [26]. This approach allowed for the design of hollow fiber reinforced endodontic posts which were recently introduced into the market [15].

Within this study, a variety of fiber posts, including hollow posts, were characterized through the three-point bending test for assessing mechanical properties such as maximum load, elastic modulus, and axial and bending stiffness. From the three-point bending results (Figure 2), it is possible to identify an initial linear portion of the load-displacement curve that allows for the detection of the elastic behavior of the posts and cement. By computing the linear best fit of the curves, the steepness of the linear region (dP/dy) is detected and the stiffness of posts and composite cement is calculated (Table 2). In particular, the steepness of carbon fiber posts filled or not with cement (HCP, HCP + MEC) is significantly higher (*p* < 0.01) than the other samples. This result can be ascribed to the carbon fiber reinforcement that increases the material stiffness and allows a more rigid response to the post itself. The significantly lower values of steepness (*p* < 0.01) are achieved by the composite cement (MEC). Observing the results for the steepness of hollow posts filled or not with cement (HCP + MEC, HGP + MEC), posts filled with cement achieve a higher stiffness but with no significant difference. The Young’s modulus measured for GP (Table 2) is consistent with values reported in the literature for glass fiber reinforced endodontic posts [27]. Little is known about the properties of hollow composite endodontic posts. Young’s moduli and the steepness of the linear region measured for HGP and HGP + MEC are similar to those recently reported in the literature but some inconsistency can be also observed [15]. This inconsistency can be ascribed to the equations used to compute the Young’s modulus and steepness of the linear region, as it is not appropriate to use the equations of compact beams to compute the properties of hollow beams.

In analyzing the load-displacement curves, it is worth noting that hollow posts (i.e., HGP and HCP) do not show a well-defined break point. This behavior may be ascribed to mechanical instability. In fact, circular tubes subjected to bending loads are prone to the ovalization of the mid-span cross section. Ovalization evolves until a critical value is reached, and from that point the hollow cylinder will buckle [28]. The ovalization of circular tubes is combined with the wrinkling phenomenon, which is the development of short wavelength periodic ripples on the compressive side of the shell. Soon after the wrinkling phenomenon occurs, the circular tube buckles. The load instability limit raises as the shell thickness increases as a direct consequence of the ovalization caused by bending loads [29]. The load-displacement behavior of HGP and HCP as the relative maximum point is reached (Figure 2) suggests that hollow posts are prone to buckling. Figure 2 clearly shows that the mechanical behavior in the elastic region of hollow posts (HGP and HCP) is not affected by the presence of the composite cement (HGP + MEC and HCP + MEC), and similar stiffness values were recorded between hollow posts and hollow posts filled with cement (Table 2). The composite cement occupies the region of the post close to the neutral axis, therefore its contribution to the post stiffness can be neglected. Instead, in the plastic region of the load-displacement curve (Figure 2), the mechanical behavior of hollow posts filled with the composite cement completely differs from that of hollow posts. The presence of the composite cement in the cavity of hollow posts prevents ovalization and buckling. As a consequence, the strength of HGP + MEC is close to that of GP (Figure 2 and Table 2).

Comparing all the investigated posts, a significantly higher rigidity (*p* < 0.01) was observed for HCP. This result is consistent with the well-known effect of carbon fiber reinforcement that provides stiffness higher than other types of reinforcement (e.g., glass fibers).

It is recognized that cyclic loading of teeth and restored teeth reduces the residual strength. Using a loading frequency of 2 Hz and a peak load of 250 N in an air environment, Mannocci et al. found that some premolars restored with carbon or quartz fibers achieved failure after about 200,000 cycles, while all specimens restored with particulate composite achieved failure at about 10,000 cycles [30]. Therefore, cyclic loading affects the residual strength of restored teeth, and the fatigue endurance limit of premolars restored with composite materials is lower than 250 N. Through our experiments, specimens were cyclic loaded at a frequency of 2 Hz and a peak load of 100 N, and no premature failure was observed up to 1 million cycles. On the other hand, using canine dental specimens restored with aesthetic glass fiber posts, the fracture bending strength after 500,000 cycles at 1.7 Hz with a peak load of 250 N in an air environment was 762.2 N [31], lower than the one (1083 N) that we detected for premolars restored with similar posts. This difference can be ascribed to the different dental specimens and loading condition.

Fracture strength of teeth after fatigue was evaluated by analyzing the maximum load achieved by the specimens. The control group (Group 1) achieved a significantly higher strength (*p* < 0.01) than the other groups, while a significantly lower strength (*p* < 0.01) was attained for the MOD group. No significant difference was found between teeth restored with different types of posts. Therefore, whatever strategy is used to restore a damaged tooth, the strength is lower than that of sound teeth (Group 1). This strength reduction can be ascribed to the stress distribution, as the endodontic post does not homogeneously distribute the load along the surrounding root walls weakened by the root canal preparation [6]. The fracture strength of specimens restored through particulate composite (Group 2) is consistent with that measured for single-rooted maxillary premolars restored with flowable and micro-hybrid resins [32]. The fracture resistance of specimens restored with glass fiber endodontic posts (Group 3) is consistent with that measured for first and second molars restored with translucent glass fiber posts [33]. The fracture strength measured for MOD cavities restored with hollow carbon fiber endodontic posts (Group 5) is consistent with strength values measured for mandibular first premolars restored with compact carbon fiber posts [34].

Failure mode and fracture patterns (Figure 4 and Figure 5) are strictly related to the axial stiffness of the post itself. An endodontic post with a high axial stiffness transfers the higher stress at the root canal walls, thus promoting an unfavorable fracture under the cement-enamel junction. This type of fracture does not allow for the repairing and reconstruction of the tooth if failure occurs. In the current research, the higher axial stiffness was achieved by the hollow carbon post presenting the higher percentage of unfavorable fractures (80% of the specimens). Besides, fiber posts with a lower axial stiffness allow for the obtaining of a lower stress concentration at the root canal walls and the promotion of a more uniform stress distribution to the coronal dentin. These posts lead to a favorable fracture that can often be repaired. Hollow glass posts that present a significantly lower axial stiffness than other posts leads to a high percentage of favorable fractures (80% of the specimens). On the other hand, the compact glass post that has a higher axial stiffness than the hollow one, and achieves a 70% of favorable fractures (Figure 4 and Figure 5). A cohesive type of fracture occurring through dentine (Figure 5a) was observed in the control group (Group 1), while an adhesive type of fracture occurring at the composite-dentine interface was found for premolars restored through the composite material (Group 2). It is worth noting that teeth restored through fiber posts (Groups 3, 4, and 5) showed a mixed type of fracture (Figure 5c–e) involving the adhesive interface between the fiber post and the cement as well as a cohesive interface occurring in both the luting cement and dentine.

Very little is known of the in vitro behaviour of restored teeth undergoing the combination of cyclic loading in a wet environment. Most of the in vitro literature investigates the fatigue behaviour of restored teeth at room temperature in an air environment [30,31]. Water itself is an important component of mineralized tissues such as dentin, and the mechanical properties of dentin depend on the moisture level, and dehydrated dentin is recognized to be more brittle than hydrated dentin, as toughness of dentin mainly depends on the organic collagen matrix and dehydrated collagen becomes brittle [35]. Although our experimental set-up considered the combination of cyclic loading in a water environment at 37 °C, further research is needed to investigate the effect of the pH on the dynamic mechanical stability of restored teeth.

## 5. Conclusions

Within the limitations of our in vitro investigation the following conclusions may be drawn:

-Fiber reinforced hollow endodontic posts allow the reverse extrusion of the luting cement, avoiding the entrapment of air bubbles.-Three-point bending tests show that the stiffness of hollow glass fiber posts filled with the cement is similar to that of compact glass fiber posts, thus suggesting that the main contribution to the post stiffness is provided by the external composite layers of the post.-The strength of premolars having an MOD cavity restored through particulate composite material is significantly lower than that of a sound tooth and that of post and core restored teeth.-Carbon fiber reinforced posts retain the maximum strength among the investigated post and core restorations, but they also lead to an unfavorable type of fracture.-The more compliant glass fiber reinforced posts allow the MOD cavity to be restored with a strength close to that of a carbon fiber post, but the type of fracture is more favorable, thus allowing for a further treatment in the case of tooth failure.

## Figures and Tables

**Figure 1 materials-15-01141-f001:**
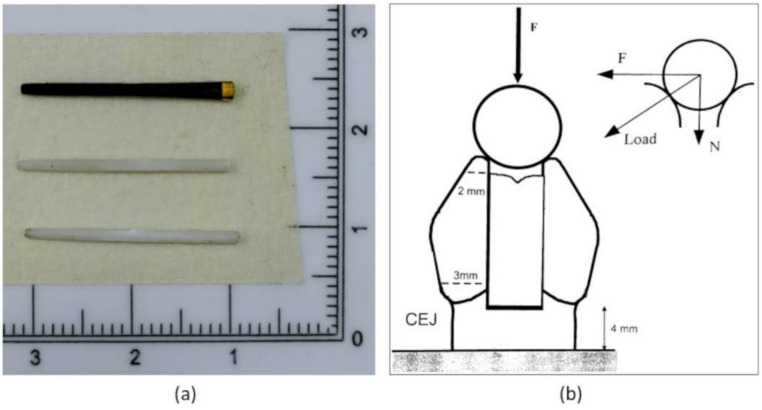
(**a**) the three types of investigated endodontic posts, (**b**) cyclic loading condition.

**Figure 2 materials-15-01141-f002:**
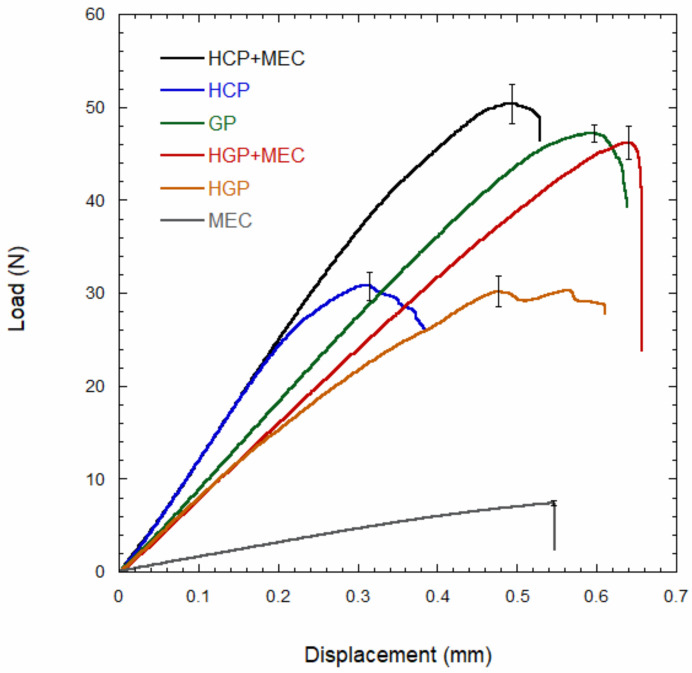
Results from three-point bending tests: load-displacement curves for hollow carbon fiber post (HCP), hollow glass fiber post (HGP), compact glass fiber post (GP), dual cement Maxcem Elite Chroma (MEC), hollow carbon fiber post filled with Maxcem Elite Chroma cement (HCP + MEC) and hollow glass fiber post filled with Maxcem Elite Chroma cement (HGP + MEC). The bars denote the standard deviation at the maximum load.

**Figure 3 materials-15-01141-f003:**
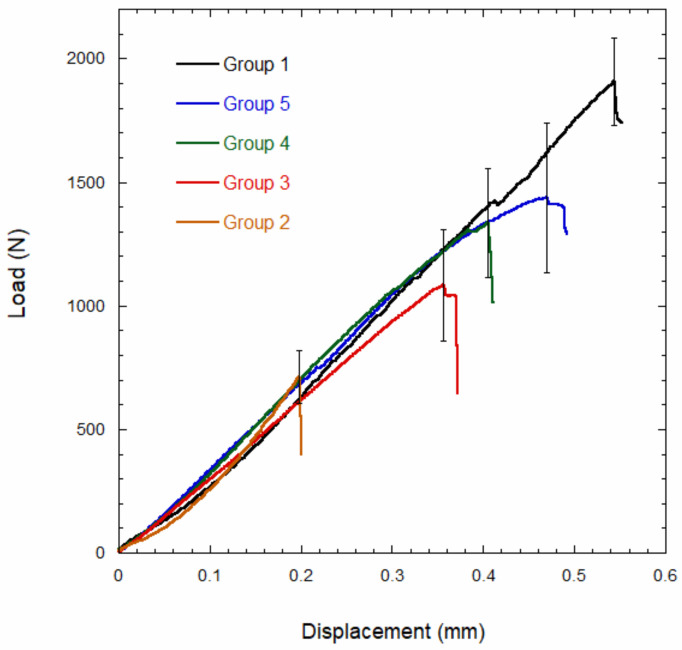
Mechanical behavior after fatigue of maxillary premolars (Group 1), premolars having an MOD restored with composite (Group 2), premolars with GF (Group 3), premolars restored with HGP (Group 4), and premolars restored with HCP (Group 5).

**Figure 4 materials-15-01141-f004:**
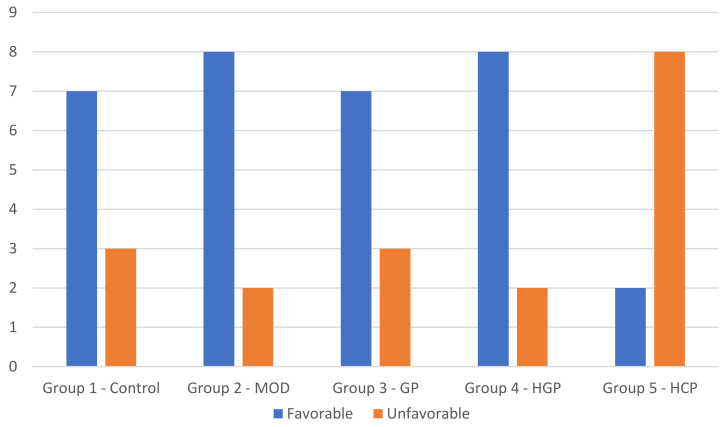
Failure mode. Number of teeth for each group characterized by favorable and unfavorable failure mode.

**Figure 5 materials-15-01141-f005:**
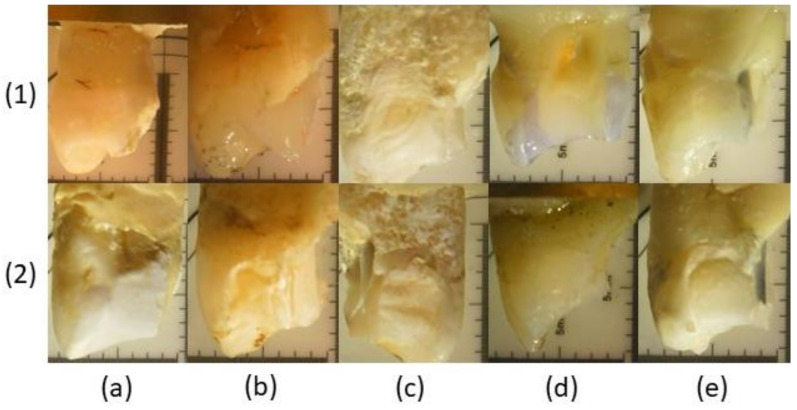
Digital microscopy to define (**1**) favorable and (**2**) unfavorable fractures of (**a**) control group, (**b**) MOD group, (**c**) glass fiber post group, (**d**) hollow glass fiber post group and (**e**) hollow carbon fiber post group.

**Figure 6 materials-15-01141-f006:**
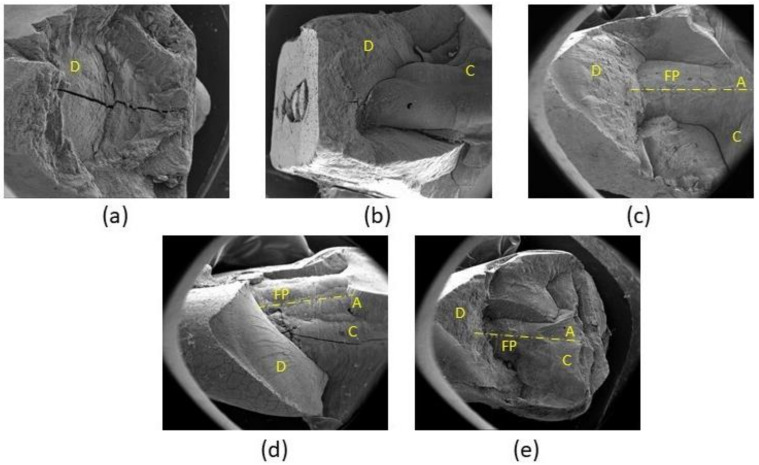
Scanning Electron Microscopy (SEM) images of (**a**) control group, (**b**) MOD group, (**c**) glass fiber post group, (**d**) hollow glass fiber post group and (**e**) hollow carbon fiber post group. D = dentine, C = composite, FP = fiber post, A = fiber post axis.

**Table 1 materials-15-01141-t001:** Composition and geometrical details of endodontic composite posts and cement. R and r represent the external and internal radius of hollow posts, respectively.

Materials	Manufacturer	Code	Composition	R (mm)	r (mm)
**Hollow Carbon Post**	Isasan (Italy)	HCP	- Carbon fibers 60% - Bisphenol-A + methyloxirane 40% - Barium sulfate traces	1.2	0.5
**Hollow Glass Post**	Isasan (Italy)	HGP	- Silica fibers 55% - Diphenylpropane + methyloxirane 45%	1.2	0.5
**Glass Post**	Isasan (Italy)	GP	- Silica fibers 55% - Diphenylpropane + methyloxirane 45%	1.2	
**Maxcem Elite Chroma**	Kerr (CA, USA)	MEC	- 2-hydroxyethyl methacrylate - 2-hydroxy-1,3-propanediyl bismethacrylate - 7,7,9 (or 7,9,9) trimethyl-4, 13-dioxo-3, 14-dioxa-5, 12-diazahexadecane-1, 16-diyl bismethacrylate - Propylidynetrimethanol, ethoxylated, esters with acrylic acid - Ytteribium trifluoride	Square cross-section 1 × 1 mm^2^	

**Table 2 materials-15-01141-t002:** Geometrical and mechanical properties of endodontic posts and cement. Numbers in brackets denote the standard deviation.

	First Moment of Area	Second Moment of Area	Maximum Load	dP/dy	E	Axial Stiffness	Bending Stiffness
	(mm^2^)	(mm^4^)	(N)	(N/mm)	(GPa)	(kN)	(kN·mm^2^)
**HCP**	0.94	0.0987	31.57 (1.52)	128.05 (2.47)	59.38 (1.14)	55.82 (1.07)	5.86 (0.11)
**HGP**	0.94	0.0987	30.48 (1.71)	83.67 (2.11)	38.80 (0.98)	36.47 (0.92)	3.83 (0.10)
**GP**	1.13	0.102	46.45 (0.92)	89.63 (2.13)	40.22 (0.95)	45.44 (1.07)	4.10 (0.10)
**MEC**	1.00	0.083	7.39 (0.27)	16.63 (1.05)	9.17 (0.58)	9.17 (0.58)	0.76 (0.05)
**HGP + MEC**	1.13	0.102	46.17 (1.78)	84.37 (3.03)	-	38.27 (0.93)	3.86 (0.11)
**HCP + MEC**	1.13	0.102	50.34 (2.10)	128.73 (1.91)	-	56.62 (1.08)	5.89 (0.11)

**Table 3 materials-15-01141-t003:** Fracture strength after fatigue of maxillary premolars (Group 1), premolars having an MOD restored with composite (Group 2), premolars restored with GF (Group 3), premolars restored with HGP (Group 4), and premolars restored with HCP (Group 5). Numbers in brackets denote the standard deviation.

	Group 1—Control	Group 2—MOD	Group 3—GP	Group 4—HGP	Group 5—HCP
**Load (N)**	1909 (177)	715 (107)	1083 (224)	1336 (221)	1467 (304)

## Data Availability

The data presented in this study are available upon request from the corresponding author.

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
