# Peer review of "Fracture Strength and Failure Modes of Endodontically Treated Premolars Restored with Compact and Hollow Composite Posts Subjected to Cyclic Fatigue"

_materials, 2022, doi:10.3390/ma15031141_

Round 1

Reviewer 1 Report

An interesting work is conducted on the mechanical behavior carbon or glass fiber rein forced endodontic posts. Mechanical properties of these posts were assessed through bending and fatigue cycling tests. It is found that composite posts increase the mechanical stability of restored premolars. The following comments can be applied to improve the quality of the paper.

* Abstract

1# The words of “background results and conclusions” in the abstract should be removed. In addition, it is suggested to provide some quantitative indicators and compare the mechanical properties of several composites.

* Introduction

1# The present introduction writing is relatively chaotic, and it only includes two paragraphs. It’s difficult to distinguish the research background, application field, research status from the others’, the unsolved problems and the present research work. It is suggested that the author considers the above factors and divide the writing of the introduction into several paragraphs.

2# In the present paper, the authors applied carbon fiber and glass fiber reinforced resin composites to the dentistry is to preserve teeth. However, the composition, properties and application fields of fiber reinforced resin composites are not introduced. In addition, when considering the actual teeth service environment, the long-term performance of fiber reinforced resin composites under fatigue loading (chewing process) and hygrothermal environment (the temperature and taste of food and water) should be summarized and analyzed in detail to simulate the performance evolution of composites inside teeth. The following work on the basic information and the fatigue properties and durability of composites in hygrothermal environment can be used for reference.

Durability: https://doi.org/10.1016/j.compstruct.2021.115060.   Durability of components of FRP-concrete bonded reinforcement systems exposed to chloride environments. Composite Structures, 2022, 279, 114697.   Fatigue properties: International Journal of Fatigue, 2020, 134: 105480.  

In addition, the introduction lacks of the necessary summary on the others’ work. The solved and unsolved problems should be further proposed.

* Materials and Methods

1# It is suggested to give the physical pictures of three kinds of endodontic posts. In addition, the relevant properties and information of carbon fiber, glass fiber and epoxy should also be further provided.

2# The three-point bending test in Section 2.2 and the bending properties in Section 2.3 should be combined together if the same content is represented.

* Results and discussion

1# In Figure 1 and table 2, the load-displacement curves are provided from three-point bending tests. Why not provide the bending strength? In addition, the curve in Figure 1 is difficult to distinguish the different conditions. It is suggested that the authors replace them with a high-definition pictures.

2# In Table 3, the fracture strength of the control after fatigue is the maximum compared with other conditions. Does this mean that the fatigue performance of maxillary premolars deteriorates after the restoration? Please provide a reasonable explanation.

3# The failure mode is figure 4, not figure 3. Please verify it.

4# For SEM, carbon fiber and glass fiber are not shown in the picture. Why? In addition, the bonding properties of fiber/resin interface should be detected after the fatigue loading from SEM.

5# In the discussion part, it is suggested that the authors quantitatively compare the present results with others’ work and present them through the figures or tables.

* Conclusions

The conclusion should be further enriched, including several key findings.

Author Response

We thank the reviewer for his valuable comments that allowed us to improve the quality of the manuscript. The improvements that have been applied are reported in the attached file

Reviewer 2 Report

The authors study mechanical properties of three types of endodontic posts, namely, a carbon fiber hollow post, a glass fiber hollow post, and a compact glass fiber post. Fatigue cycling and the static strength tests of restored teeth were performed. Failure modes are discussed and corresponding recommendations are proposed. The work is of high scientific and practical quality. However, the authors should correct a few shortcomings to make the manuscript acceptable for publication.

(1) The aim of the current study was to evaluate the fracture resistance and fracture patterns of endodontically treated maxillary premolars subjected to mesial-occlusal-distal (MOD) cavities restored using a resin composite core with different types of fiber posts under cyclic loading. Could the references be provided showing an effect of cyclic loading on residual strength of teeth? For example: Mannocci F, Ferrari M, Watson T. Intermittent loading of teeth restored using quartz fiber, carbon-quartz fiber and zirconium dioxide ceramic root canal posts. J Adhes Dent. 1999;1:153-8.

(2) Lines 200-202: The sentence “Then, specimens were subjected to fatigue stress cyclic loading, in a water environment at a constant temperature of 37°C, for one million cycles with a sinusoidal variable loading in the range 10-100 N at a frequency of 2 Hz” describes a technique of applying cyclic loading to the specimens. However, it is not explained whether the loading was compressive or of another kind. If so, I propose to replace a too wordy phrase “fatigue stress cyclic loading” with “cyclic compressive loading”.

(3) Could the authors substantiate the set cyclic loading range 10-100 N (probably, in relation to an average value of the ultimate static compressive load)? Corresponding references may be added. For example: https://doi.org/10.1590/S1678-77572006000400016

(4) Could the references be provided showing a significant/insignificant difference in effects of a water environment (this study) and the oral environment with corresponding pH on mechanical behavior of endodontically treated teeth under cyclic loading? Please comment this case.

(5) In Fig.1, please replace “,” (comma) with “.” (dot) in the numbers corresponding to the horizontal axis.

(6) Fig.5, for each case (c, d, e) the fiber post axis direction need to be indicated either by an arrow or otherwise.

Author Response

We thank the reviewer for his valuable comments that allowed us to improve the quality of the manuscript. The  improvements that have been applied are reported in the attached file.

Round 2

Reviewer 1 Report

Although the authors try to make a modification to the paper, the overall modification is not enough. Some specific statements should be addressed to improve the overall quality of paper again.

  • Although the present introduction part is divided into several paragraphs, the overall logical relationship is still chaotic, and there is no clear relationship on the research background, research results and present research work. In addition, the innovation of present research work should also be further highlighted.
  • The author added the following sentences to the revised paper: “Endodontic composite posts mainly consists of straight fibers (i.e. glass or carbon fibers) embedded into an epoxy matrix. The stability of the fiber-matrix interface in a wet environment largely depends on the quality of this interface, and it is reported that, in chloride solution, the interfacial damage between fibres and matrix is delayed as a smooth-surfaced composite is used”. The summary on the degradation of composites in hygrothermal environment is one-sided and it is not only dependent on the fiber/matrix interface. As known, the hydrolysis and plastication of resin matrix are also another main factor causing the mechanical and thermal degradation of composites. Therefore, the relevant research findings should also be summarized. In addition, the composition, properties and application fields of fiber reinforced polymer composites should be addressed.
  • The conclusion should be concise, including the main findings with three to four points. Further improvement is recommended.

Author Response

We thank the reviewer for the valuable comments that allowed us to further improve the quality of the manuscript.

The introduction section has been implemented and restructured into seven paragraph. The first paragraph dealing with clinical problems related to damaged premolars, the second paragraph introducing continuous fiber reinforced endodontic posts, the third paragraph considering the cementation of pre-fabricated posts in the root canal, the fourth paragraph introducing the novel hollow composite post very recently introduced on the market, the fifth paragraph considering the contemporary clinical and in vitro concerns related to the post and core restoration, the six paragraph has been implemented as reported below, the seventh paragraph reporting the aim of our investigation.

Six paragraph: Endodontic composite posts mainly consists of straight fibers (i.e. glass or carbon fibers) embedded into an epoxy matrix. The stability of the fiber-matrix interface in a wet environment largely depends on the quality of this interface, and it is reported that, in chloride solution, the interfacial damage between fibres and matrix is delayed as a smooth-surfaced composite is used [19]. On the other hand, debonding of the fiber/resin interface in rods made of carbon/glass fiber reinforced epoxy has been observed, and it has been ascribed to water diffusion that hydrolyses and plasticizes the resin matrix weakening the bond between fiber and resin [Ref 20: Li, C., Yin, X., Liu, Y., Guo, R., & Xian, G. (2020). Long-term service evaluation of a pultruded carbon/glass hybrid rod exposed to elevated temperature, hydraulic pressure and fatigue load coupling. International Journal of Fatigue, 134, 105480]. Therefore, it is very important to replicate the complex oral environment for assessing in vitro the biomechanical behaviour of applied composite endodontic posts. Little is known on the in vitro behavior of premolars restored with endodontic posts stressed through cyclic loading in a wet environment at 37°C.

The Conclusion section has been restructured as follows:

Within the limitations of our in vitro investigation the following conclusions may be drawn:

-             Fiber reinforced hollow endodontic posts allow the reverse extrusion of the luting cement, avoiding the entrapment of air bubbles.

-             Three-point bending tests show that the stiffness of hollow glass fiber posts filled with the cement is similar to that of compact glass fiber posts, thus sug-gesting that the main contribution to the post stiffness is provided by the ex-ternal composite layers of the post.

-             The strength of premolars having an MOD cavity restored through particulate composite material is significantly lower than that of a sound tooth and that of post and core restored tooth.

-             Carbon fiber reinforced post retains the maximum strength among the investigated post and core restorations, but they also lead to an unfavorable type of fracture.

-             The more compliant glass fiber reinforced posts allows to restore MOD cavity with a strength close to that of carbon fiber post, but the type of fracture is more favorable, thus allowing for a further tooth re-treatment in the case of tooth failure.
